# Effects of Coated Separator Surface Morphology on Electrolyte Interfacial Wettability and Corresponding Li–Ion Battery Performance

**DOI:** 10.3390/polym12010117

**Published:** 2020-01-05

**Authors:** Ruijie Xu, Henghui Huang, Ziqin Tian, Jiayi Xie, Caihong Lei

**Affiliations:** Guangdong Provincial Key Laboratory of Functional Soft Condensed Matter, Guangdong Provincial Engineering Laboratory of Energy Storage Materials and Devices, School of Materials and Energy, Guangdong University of Technology, Guangzhou 510006, China; 2111702033@mail2.gdut.edu.cn (H.H.); tianziqin@outlook.com (Z.T.); xiaojy710@gmail.com (J.X.)

**Keywords:** interfacial wettability, immersion free energy, Li–ion battery, coated separator

## Abstract

In order to study the effect of interfacial wettability of separator on electrochemical properties for lithium–ion batteries, two different kinds of polyvinylidene fluoride-hexafluoropropylene (PVDF–HFP) solution are prepared and used to coat onto a polypropylene (PP) microporous membrane. It is found that the cell performance of a coated separator using aqueous slurry (WPS) is better than that of the coated separator using acetone (APS) as the solvent. The separator with flat and pyknotic surface (PP and APS) has a strong polar action with the electrolyte, where the polar part is more than 80%. To the contrary, the WPS has a roughness surface and when the PVDF–HFP particles accumulate loose, it makes the apolar part plays a dominate role in surface free energy; the dispersive energy reaches to 40.17 mJ m^−2^. The WPS has the lowest immersion free energy, 31.9 mJ m^−2^ with the electrolyte, and this will accelerate electrolyte infiltration to the separator. The loose particle accumulation also increases the electrolyte weight uptake and interfacial wettability velocity, which plays a crucial role in improving the cell performance such as the ionic conductivity, discharge capacity and the C-rate capability.

## 1. Introduction

With the material science progressing, the properties of lithium–ion batteries (LIBs) have satisfied in many high demand areas, such as HEV or EV [1]. But there are still many challenges on LIBs technologies, especially for the demand of fast charging rate. For this fast charging rate, the charge time will sharply be short. The diffusion resistance of lithium ions shuttling between electrode material is the main fact which restricts the fast chargeability of LIBs [2,3]. Till now, many researchers focused on improving the ions’ diffusion velocity in electrode materials. The diffusion resistances’ route from the separator and electrolyte are seldom paid attention. In fact, this part of the resistance has a greater effect on the rapid conduction of lithium ions. Recently, Xie et al. [4] provided that the hydrophilicity of the electrolyte would help to improve the discharge capacities and cycling stability. But the interface between the separator and electrolyte are seldom given any attention.

Since the LIBs appeared in the 1990s, polyolefin materials became the first choice for the separator, such as polyethylene (PE) and polypropylene (PP). Due to its hydrophobic surface with low surface energy and bad electrolyte wettability, polyolefin separators have many disadvantages in battery application, such as poor liquid electrolyte absorption and retention, and big internal resistance. Many new functional separators are needed for high performance cells [5].

During the past few years, many efforts have been done to coating the polar functional layer onto polyolefin separators. The surface polar layer will increase the surface polarity and the wettability of electrolyte onto the separator. The first method is dissolution of the polar polymer material into the organic solvent, and then coating onto the polyolefin separator to form the microporous layer by the nonsolvent-induced phase separation (NIPS) method. The preferred polymer material is PVDF or PVDF-HFP due to its wide use in electrode material pasting. The cell using the PVDF-coated separator showed good cycle life stability and excellent rate performance at room temperature [6]. This improvement mainly comes from the surface polarity-induced electrolyte absorption increment, which is conducive to improve ionic conductivity and reduce the inner resistance [7,8]. But it has to be pointed out that the NIPS layer is usually a pyknic film-shape, which will inevitably block the penetration of liquid electrolyte into the pores of the microporous membrane. On the other hand, plenty of works focus on environmental aqueous slurry-coated separators. The aqueous slurry usually contains micron-sized powders such as alumina (Al_2_O_3_), titania (TiO_2_), silica oxide, boehmite, etc. [9,10,11,12], and aqueous polymer solutions which act as a binder, such as polyvinyl acetate (PVA). In our recent work, the PVDF–HFP powder is a new type potential coating doped powder to replace the common ceramic powders [8]. Since the ceramic powders are bound onto the microporous membrane surface, the coating surface is a roughness and polyporous structure. This polyporous structure not only promotes the ion conduction, but also resists the heat shock. Meanwhile, the capillary action from the roughness surface changed the surface dispersion force. It is obvious that the aqueous slurry-coated separators show different infiltration mechanisms than oily coating.

It is clear that two types of polar coating layer will help the electrolyte uptake increment. Continuous electrolytes on either side of the separator help to transfer the lithium ions. The wetting gap between the separator and electrolyte will form very huge inner resistance and block Li^+^ ion transportation. At this moment, the spread of the electrolyte on the separator caused by the interface interaction will greatly affect the performance of the battery, such as higher discharge capacity, rate capability and better low temperature performance. In this work, we prepared PVDF–HFP aqueous slurry-coated separators and an oily-coated separator. In aqueous slurry, the PVDF–HFP polymer powders are directly dispersed in the PVA water solution. The oily slurry is dissolved into acetone. These two types of separators’ surface energy are characterized, and the corresponding membrane and cell properties are also obtained. The target of this work is to clarify the relationship between membrane surface characterization and its effect on corresponding cell performance.

## 2. Experimental

### 2.1. Materials

The PP microporous membrane with the thickness of 16 μm is supplied by Shenzhen Senior Materials Company (Shenzhen, China). According to the supplier, the membrane has a porosity of 45%, and a Gurley value (characterizing the air permeability, the lower the Gurley value, the better the air permeability) of 240 s/100 mL. PVDF–HFP with the HFP content of 12% was obtained from Arkema (Colombes, France). Acetone was obtained from FuYu Fine Chemical Co. Ltd. (Tianjin, China). PVA with a saponification degree of 80% was supplied from Shenzhen Huasu Technology Company (Shenzhen, China).

### 2.2. Coated Membrane Preparation

The water-based PVDF–HFP solution was prepared using PVDF–HFP as the doped powder and PVA as the hydrophilic polymer binder. Firstly, PVA was completely dissolved in the deionized water by stirring for 1 h at 70 °C. The PVDF–HFP particles (120 g) were also dispersed in 1 L of deionized water with a mechanical stirrer for 1 h under room temperature. Then, the prepared PVA solution was added into the PVDF–HFP dispersion and additionally 1 h stirring was carried out. For comparison, the acetone-based PVDF–HFP solution was also prepared by dissolving PVDF–HFP in acetone with a mechanical stirrer for 1 h under room temperature.

The resulting slurry was coated onto both sides of a PP microporous membrane via a dip-coating process. The coated membrane was dried in a vacuum oven at 60 °C for more than 24 h. The total thickness of coated membranes was kept at less than 20 μm. The water-based PVDF–HFP solution-coated separator is denoted as WPS, while the acetone-based PVDF–HFP solution-coated separator is denoted as APS. The initial PP separator is denoted as PP.

### 2.3. Characterization of PVDF-HFP Coated Separators

Surface and cross-section morphology of the coated separators were characterized by scanning electron microscopy (SEM; S3400N and SU8010, Hitachi, Tokyo, Japan). All the samples were sputtered with a platinum ion beam for 300 s before test. The acceleration voltages used in the SEM experiment is 10 KV. The pore size and coating layer thickness are calculated by the software of ImageJ.

The air permeability of coated samples was characterized by a Gurley Densometer model No.4150 (Gurley Precision Instruments, Troy, NY, USA) according to ASTM D726.

In order to measure the electrolyte uptake, the coated membrane was immersed in 1.0 M LiPF6 in EC/DMC for 1 h. Then, the membrane was taken out from the electrolyte solution and excess electrolyte solution on the surface of membrane was removed by wiping with filter paper. The uptake of electrolyte solution was determined using the following relationship:(1)$$uptake\left(\%\right)=\left(W-W_{0}\right)/W_{0}\times 100\%$$
here, *W*_0_ and *W* are the weights of the membrane before and after soaking in the liquid electrolyte, respectively.

The electrolyte contact angle was determined by means of the sessile drop method using a contact angle meter G-1 model (ERMA Inc., Tokyo, Japan). The electrolyte droplet was limited to about 0.5 μL to prevent gravitational distortion of its spherical profile.

The dynamic contact angle analysis test (DCAT21, Dataphysics, Filderstadt, Germany) was used to determine wettability about the PP separator and coated separator. The DCAT experiments were performed at a stage motor speed of 0.05 mm s^−1^ (surf. detection), 0.01 mm/s (measurement adv.), 0.20 mm s^−1^ (measurement rec.) with a surface detection threshold of 0.15 mg and an immersion depth of 3 mm, respectively.

Surface energies are calculated from a dynamic contact angle analysis test result. For this purpose, two liquids with different surface tension are used, water and ethylene glycol. Table 1 gives the surface tension components of liquids reported by van Oss [13] used in this work.

Here, we use the OWK method to calculate the solid surface free energy, γ_S_, calculation from the equilibrium liquid contact angles [14]. According to the Fowkes theory [15] that interactions between nonpolar solid and liquid can only be attributed to dispersion forces, Owens-Wendt-Kaelbel [16] suggested that dispersive (d) and polar (p) forces may be important across the interface for polar solids (S) and liquids (L). Considering a geometric mean relationship, the following equation is recommended for γ_S_ calculation: (2)$$\gamma_{S}=\gamma_{S}^{d}+\gamma_{S}^{p}$$
(3)$$W_{A}=\left(1+\cos\theta \right)\gamma_{L}=2\sqrt{\gamma_{S}^{d}\gamma_{L}^{d}}+2\sqrt{\gamma_{S}^{p}\gamma_{L}^{p}}$$
here *W*_A_ is a work of adhesion, *γ*_L_ is the liquid surface tension, *γ*_S_ is the solid surface tension and *γ*_d_ and *γ*_p_ are the dispersive and polar components of surface free energy. To obtain the surface energy *γ*_S_ value, the contact angles *θ*_1_ and *θ*_2_ of separators to water and ethylene glycol are obtained by the dynamic contact angle analysis test.
(4)$$\left\{ \begin{array}{l} \left(1+\cos\theta_{1}\right)\gamma_{L1}=2\sqrt{\gamma_{S}^{d}\gamma_{L1}^{d}}+2\sqrt{\gamma_{S}^{p}\gamma_{L1}^{p}}\\ \left(1+\cos\theta_{2}\right)\gamma_{L2}=2\sqrt{\gamma_{S}^{d}\gamma_{L2}^{d}}+2\sqrt{\gamma_{S}^{p}\gamma_{L2}^{p}} \end{array}\right.$$

The dispersive and polar components of surface free energy can be calculated according to Equation (4).

### 2.4. Electrochemical Measurements of PVDF-HFP Coated Separators

The charge/discharge and cycling tests of the LiFePO_4_ cathode using the PVDF–HFP-coated membrane and the PP microporous membrane as the separators, which are saturated with the liquid electrolyte, were conducted by assembling a coin-type cell with lithium metal foil as the counter and reference electrode. The used cycler was a Land tester (CT2001A), and the voltage was between 2.7 and 4.2 V at a current density of 0.2 C, based on the LiFePO_4_ working electrode. The working electrode was prepared by coating the N-methyl-1-pyrrolidone-based slurry containing LiFePO_4_ (STL Energy Technology Co., Ltd., Tianjin, China), acetylene black and PVDF (Binder, Tuttlingen, Germany) in a weight ratio of 8:1:1 on an aluminum foil (thickness: 20 μm) using a doctor-blade technique, and the cast foils were then punched into circular pieces (*d* = 15 mm) and dried at 120 °C for 12 h under vacuum. All cells were assembled in an *Ar*-filled glove box.

The electrochemical stability was determined by performing linear sweep voltammetry at 5 mVs^−1^ using stainless steel and Li foil as the respective working and counter electrodes. The ionic conductivity was calculated as in Cao’s work [17], while the MacMullin number (*N*_m_) can be calculated as in Zhai’s work [18]. In the electrochemical impedance spectroscopy (EIS) measurements, the excitation voltage applied to the cells was 5 mV and the frequency range was between 300 kHz and 0.05 Hz. The rate capability was evaluated by charging–discharging at various current densities (0.1 C, 0.5 C, 1 C, 2 C and 5 C) for 5 cycles each, continuously.

## 3. Results and Discussion

The morphology of PP, WPS and APS were investigated using SEM and shown in Figure 1. The uniform distributed pores with diameter in a range of 20–200 nm can be observed on the surface of the PP microporous membrane, as seen in Figure 1a. The coated separators show two types of surface structure. APS presents a kind of porous structure with pore sizes ranging from 1 to 6 μm in the PVDF–HFP copolymer layer, which can be attributed to the evaporation of acetone during the drying of the coated membrane, as shown in Figure 1b. But in the corresponding cross-section, the APS coating layer (Figure 1c) shows film-shaped structure that is closely connecting with the PP microporous membrane. Figure 1d shows that the WPS has many voids among the PVDF–HFP powders. The base membrane can still be observed on the coated surface. In the corresponding cross-section, Figure 1e, there are also voids between the coating layer and the matrix membrane. This kind of porous structure is expected to play an important role in improving the liquid electrolyte wettability velocity and ionic conduction. This unpyknotic coating layer has the benefit on increasing the electrolyte uptake and retention.

Figure 2 gives the electrolyte absorption versus time curves for PP, APS and WPS. Compared with the PP membrane, the coating layer induces a pronounced increase of electrolyte absorption, retention amount and absorption rate [19]. This change shows that the affinity between the separator and the liquid electrolyte is markedly enhanced. Even though the surface material of WPS and APS are all the PVDF–HFP, they have the similar surface properties but different electrolyte weight uptake velocity. The roughness surface has more benefit on electrolyte uptake velocity.

The Gurley value is one of the most important technical parameters for microporous separator. The Gurley values of PP, APS and WPS are 240, 360 and 282 s·100 mL^−1^, respectively. It can be seen that the existence of coating layer destroys the air permeability. Especially, the coating using the acetone-based PVDF–HFP copolymer solution results in the increase of the Gurley value by 50%. Compared with APS, the WPS shows a lower Gurley value. The voids between the coating layer and the base membrane in Figure 2 afford the passage for air pass-through.

The ionic conductivity is a significant parameter for the electrochemical performance of the battery. The specific ionic conductivity calculated from *R*_b_ at room temperature is listed in Table 2. The order of the ionic conductivity is WPS (0.73 × 10^−3^ S cm^−1^) > APS (0.59 × 10^−3^ S cm^−1^) > PP membrane (0.42 × 10^−3^ S cm^−1^). The ionic conductivity increases with the electrolyte uptake due to the availability of more Li–ions for conduction in the same volume.

The *N*_m_ is related to ionic conductivity. The *N*_m_ for PP, APS and WPS are 20.7, 14.7 and 11.9, respectively. The value of the coated membrane is lower than that of PP, indicating the higher electrolyte uptake and wettability have the benefit on the battery performance. Compared with APS, WPS shows higher electrolyte uptake and better wettability, leading to lower *N*_m_.

The discharge voltage profiles of the cells using different separators at different current rates are shown in Figure 3. It can be clearly seen that the discharge voltage plateau drops with increasing the current rate for all cells. The WPS shows a better capacity as compared to the battery using APS and PP. The improved rate performance is ascribed to the favorable interfacial charge transport between the electrodes and the electrolytes in the cell, because the coating layer on both sides of the separator is able to assist in the adhering of the separator to the electrodes after soaking in the electrolyte. However, the discharge capacity of the APS shows not much difference with PP. Although this kind of coated separator shows higher electrolyte uptake and better wettability as shown in Figure 2, the film-shaped structure is observed in the coating layer. The paths that supply to transport the lithium ion may be shut down by the coating layer. Compared with APS, WPS exhibits higher electrolyte uptake and better wettability. In addition to this, some pore structure is formed in the coating layer. All these are beneficial to the improvement of rate performance.

Figure 4 shows the discharge capacities of lithium–ion batteries assembled with various separators during experiments in which the C rate was increased every five cycles within the range of 0.1~5.0 C. The cells with PP, APS and WPS membrane showed a capacity of 143.1, 142.5 and 145.5 m·Ah g^−1^ at 0.1 C respectively, the capacity retention ratios are about 70.2%, 70.9% and 71.2% at 2 C, and then decrease rapidly to 74.2, 76 and 81.3 m·Ah g^−1^ at 5 C, respectively. It can be seen that the discharge capacities of the cells with PP separator and APS are similar, whereas the WPS shows better performance at the high C rate. This superior rate capability can be ascribed not only to the higher ion conductivity in the PVDF–PVA layer, but also to lower interfacial resistance due to better interfacial contacts between the PVDF–PVA separator and electrodes in the lithium–ion-assembled cell [20]. 99.7% of the initial discharge capacity at a rate of 0.1 C after high current rate cycles is recovered for the batteries with PVDF separators, indicating good cycling stability. The cell using APS separator becomes worse than others, because some pores are blocked in coating process and slow down the transportation of the Li^+^.

Figure 5 shows the EIS spectra of the electrodes in the frequency range from 0.05 Hz to 300 KHz of the Li–ion cell with different separators. The impedance plots are composed of a depressed semicircle, which corresponds to the charge transfer resistance (*R*_ct_) and a linear tail at low frequency. The intercept at axis Z’, corresponding to the combination resistance *R*_e_, associates with the ionic conductivity resistance [21].

In our recent work, the suitable equivalent circuit model is obtained for the Li–ion cell with coating membrane [8]. From the equivalent circuit model (Figure 6), the electrode current could be divided into two parts in parallel: faradic current, which is caused by the electrochemical reaction; non-faradic current, which is elicited by the charge–discharge of the electrical double layer. The impedance produced by the faradic current is called faradic impedance, which could be further divided into electrochemical reaction impedance (*R*_ct_) and concentration polarization impedance. The impedance and constant phase angle element for characterizing SEI are also included in the equivalent circuit. The Warburg impedance should be excluded in the parallel circuit. Based on the equivalent circuit, the impedance parameters’ data are fitted by the software Zview. The result shows that the coating layer has the benefit on reducing the *R*_Ω_ and *R*_ct_. For the initial PP membrane, the *R*_ct_ and R_Ω_ are 97.9 Ω and 22.9 Ω, respectively. The *R*_ct_ and R_Ω_ of the cell assembled with APS are 80.1 Ω and 16.7 Ω, respectively.

The *R*_ct_ is reduced sharply to 32.0 Ω, which is attributed to the better wetting properties [22] and the better adhesion property of the PVDF–HFP coating layer towards the cathode [23]. The voids formed at the coating layer and between the coating layer and base membrane give more ion tunnel for the electrochemical reaction. But the APS shows small *R*_SEI_ 4.3 Ω than the other samples, which are due to the film-shaped coating layer in the APS, which is restrained the SEI layer formation between the electrolyte and separator. In generally, the cell with WPS shows more excellent performance.

From the separator and the corresponding cell properties, we notice that although the surface coating layer in WPS and APS are both the PVDF–HFP, the cell with the WPS separator shows more excellent property. It seems that surface polarity is not the single effect factor for electrochemical properties increasing. So we followed the surface energy parameters of the three separator samples.

The wettability of the separator is evaluated by the liquid electrolyte contact angle. Generally, a smaller contact angle corresponds to a higher affinity [24]. The contact angle of the PP membrane is 103.6 ± 0.50°, whereas it is only 15 ± 1.26° for APS and 0 ± 0.13° for WPS. Compared with APS, the WPS shows a lower static contact angle. This enhancement is attributed to three factors. The first is the film-like coating layer in APS, limiting the liquid electrolyte diffuse. The second is the rough surface of the coating layer, which allows the liquid electrolyte to infiltrate through the well-connected interstitial voids, possibly driven by the capillary force [25]. The third is that the PVDF particle and PVA have a similarity chemical polarize of the coating organic materials and the solvents [26].

The static wetting behavior has proven that the aqueous coating layer has better affinity with electrolyte. To verify the surface polarity’s contribution, the dynamic contact angle analysis test has been used.

It is well known that there is no exact algorithm which exists for calculating the surface energy of a solid surface from the contact angle data. A number of semi-empirical methods are in use for evaluation of the surface energy of a solid from wetting measurements. Most of these methods require contact angle measurements with a set of liquids of different surface tensions and polarities. It is well known that the surface tensions can be separated into a polar and a dispersive part [27,28]. Hereby, the dispersive part represents the interactions due to dispersion forces and the polar part subsumes the interactions, which are evoked by polar functional groups. The dispersive and the polar part add up to the total surface energy.

Based on the Equations (2)–(4) (in the Experimental Section), the polar and dispersive part of separators surface free energy can be calculated. The total surface free energy is the sum of two parts, and all results are listed in Table 3.

The result shows that ASP has similar surface properties as the initial PP membrane. The surface polarity of APS is not obviously increasing even after coating a PVDF–HFP layer. The flat and pyknotic layer reduce the surface roughness and induces the dispersive component turns small. But the WPS show very high dispersive surface tension which comes from the PVDF–HFP particles random accumulation. The polar induces interaction between the electrolyte and surface turns very weak by this time.

The property of electrolyte spread out on the surface of the separator is one of the most important factors to the internal resistance of the cell. Besides several parameters of the pole piece smoothness, and the component of the electrolyte, another very important influencing factor is the physicochemical compatibility of the separator surface and the electrolyte; its influence can be quantified in the free energy of immersion. The free energy of immersion Δ*G*_i_ is defined as the difference of the interfacial energy between the separator surface and electrolyte, *γ*_S_, and the surface energy of the separator [29]. This definition of the free energy of immersion can be illustrated as the comparison of the energies of the non-wetted and the wetted state of the separator.

If Δ*G*_i_ has a negative value, then the wetting of the separator surface by the electrolyte is thermodynamically favored. Positive values of Δ*G*_i_ show that the interface between separator surface and electrolyte is inconsecutive, and the system shows a higher interface resistance. Therefore, the free energy of immersion can be considered as a predictor for the contribution of thermodynamics to the electrolytic soaking. It is above mentioned how the surface energy of initial and coating separator surface can be determined, but the direct measurement of the interfacial energy γ_SL_ is not possible. In combination with Young’s equation, the free energy of immersion can be written as $\Delta G_{i}=-\gamma_{L}\cos\theta$, due to the fact that the contact angles of electrolyte on separator surfaces are not accessible for direct measurements, and then the calculation can further modified as $\Delta G_{i}=\gamma_{L}-2\left(\sqrt{\gamma_{S}^{d}\gamma_{L}^{d}}+\sqrt{\gamma_{S}^{p}\gamma_{L}^{p}}\right)$.

With the help of this equation the calculation of the free energy of immersion is possible for all combinations of electrolyte and separator surfaces, by the knowledge of disperse and polar parts of their surface energies (see Table 3). We use propylene carbonate as the typical representative of the electrolyte. The values of ΔGi derived from the mean surface energy values of the Wilhelmy wetting experiments. The free energy of immersion for the combinations of electrolyte and three separator surfaces are 13.2, 14.3 and 31.9 mJ m^−2^. It has to be emphasized that the free energy of immersion reflects hereby only the thermodynamic contribution of the wetting of the separator surface by the electrolyte, and delivers one important parameter for the electrolyte wetting process. Comparing with three types separator, the free energy of immersion shows a negative value. It means the electrolyte is easily wetting on the separator surface. The WPS sample shows the lowest Δ*G*_i_ value, and it means that the electrolyte will be dispersed onto the separator much easier. This result gives us new cognition that increaing the separator surface roughness will be more useful to electrolyte wetting rather than the surface polarity.

At the same time, the lowest Δ*G*_i_ of WPS enables the electrolyte to permeate into the separator very quickly. The interface resistance between separator and anode or cathode will obviously reduce due to the electrolyte spreads in the interface void. The body resistance will be reduced by more ionic pathways which are contributed. It is obvious that the better wettability has the benefit on reducing cell inner resistance.

## 4. Conclusions

In this paper, two types of PVDF–HFP-coated separators were fabricated and the properties of the separator and the corresponding cells were characterized. The result shows that two types of coated layer morphology are obtained. The APS sample using acetone as the solvent shows pyknotic film-shaped structure, whereas the WPS sample using water gives a typical structure showing a lot of voids on the surface and between the coating layer and base membrane. The WPS membrane shows lower contract angle and higher liquid electrolyte weight uptakes than PP separator and APS. Due to the highest electrolyte uptakes and excellent electrolyte wettability, the WPS shows the lowest bulk resistance of 1.77 Ω and smallest MacMullin number of 11.9. The ionic conductivity is 0.73 × 10^−4^ S·cm^−1^, well above that of the PP separator and APS. The cell assembled with WPS shows the longest discharge plateau at the 5 C current rate. The charge transfer resistance of the cell is only 197.1 Ω, and the conductivity is 1.92 × 10^−4^ S·cm^−1^. The separator surface properties have strongly affected the electrochemical properties. The separator with flat and pyknotic surface (PP and APS) has a strong polar action with the electrolyte, the polar part is more than 80%.

To the contrary, the WPS has a roughness surface and the PVDF–HFP particle accumulates loose, where it makes the apolar part play a dominate role in the surface free energy, and the dispersive energy reaches to 40.17 mJ m^−2^. The WPS has the lowest immersion free energy, −31.9 mJ m^−2^ with the electrolyte, and this will accelerate electrolyte infiltration to separator. The electrochemical and separator surface properties show that the WPS is suitable to act as a lithium–ion battery separator and the water-based PVDF coating slurry affords a new way to realize an environmentally friendly organic coating.

## Figures and Tables

**Figure 1 polymers-12-00117-f001:**
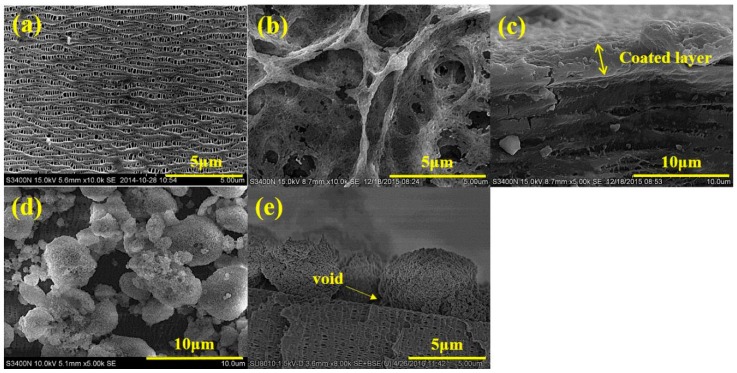
Surface morphology of the polypropylene (PP) membrane (**a**), surface morphology of acetone (APS) (**b**) and aqueous slurry (WPS) (**d**), and the sectional morphology APS (**c**) and WPS (**e**).

**Figure 2 polymers-12-00117-f002:**
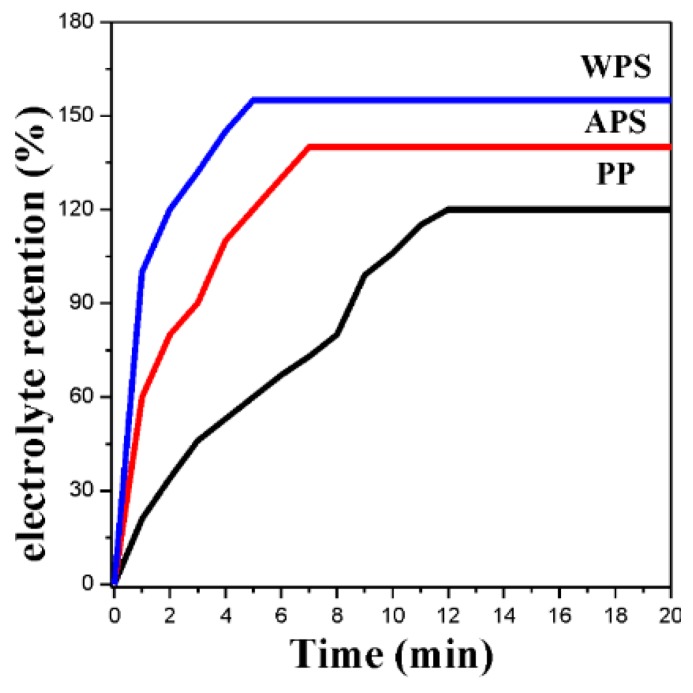
Electrolyte weight uptakes vs. time curves for PP membrane, APS and WPS.

**Figure 3 polymers-12-00117-f003:**
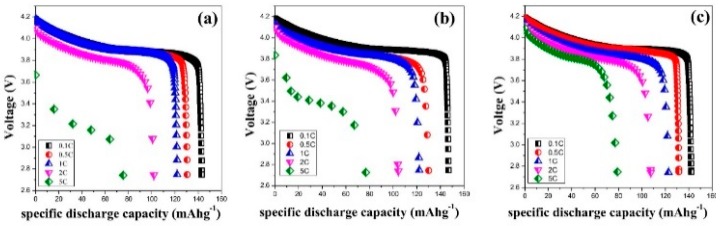
The discharge voltage profiles of the cells using different separators at different current rates, PP (**a**), APS (**b**) and WPS (**c**).

**Figure 4 polymers-12-00117-f004:**
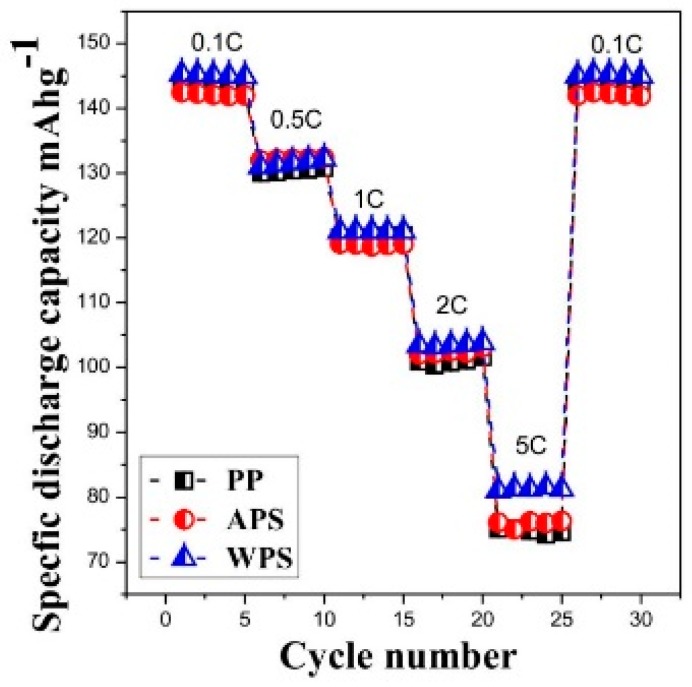
Discharge capacities of the lithium–ion batteries assembled with various separators as a function of C rate.

**Figure 5 polymers-12-00117-f005:**
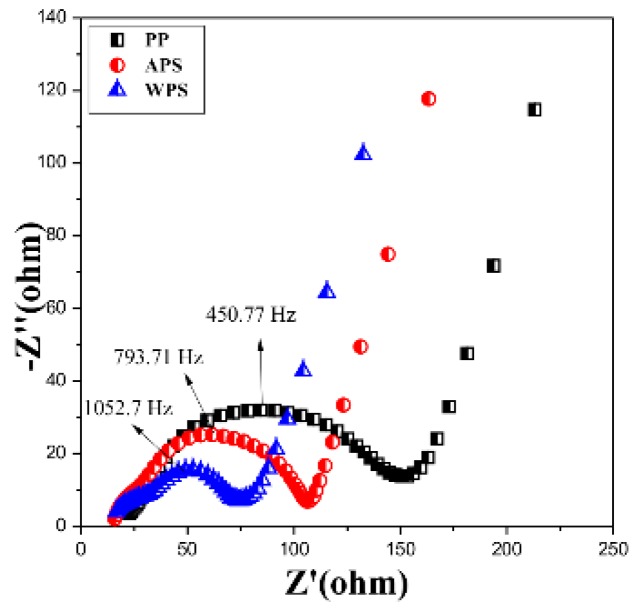
Electrochemical impedance spectroscopy (EIS) spectra of assembled battery with PP, APS and WPS as separators at room temperature.

**Figure 6 polymers-12-00117-f006:**
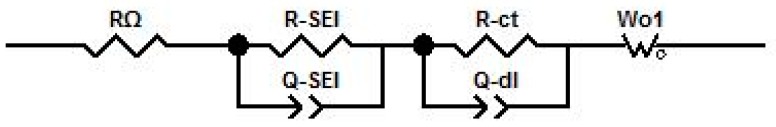
The equivalent circuit model.

**Table 1 polymers-12-00117-t001:** The surface tension components of liquids.

Liquid	γ_L_ (mJ m^−2^)	γ_d_ (mJ m^−2^)	γ_p_ (mJ m^−2^)
ethylene glycol	48	29	19
water	72.8	21.8	51
propylene carbonate	40.7	26.5	14.2

*γ*_L_ is the liquid surface tension. *γ*_d_ and *γ*_p_ are the dispersive and polar components of surface free energy.

**Table 2 polymers-12-00117-t002:** Ionic conductivity and MacMullin numbers of PP, APS and WPS.

	σ (×10^−3^ s cm^−1^)	Nm
PP	0.42	20.7
APS	0.59	14.7
WPS	0.73	11.9

**Table 3 polymers-12-00117-t003:** Surface energy values of different separators.

	*γ*_S_^d^ mJ m^−2^	*γ*_S_^p^ mJ m^−2^	*γ*_S_ mJ m^−2^	*P* %
PP	3.03	22.86	25.89	88.3
APS	4.23	20.24	24.27	83.4
WPS	40.17	0.97	41.14	2.4

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
