# Peer review of "Effects of Coated Separator Surface Morphology on Electrolyte Interfacial Wettability and Corresponding Li–Ion Battery Performance"

_polymers, 2020, doi:10.3390/polym12010117_

Round 1

Reviewer 1 Report

Generalny, papier is well design and written but some aspects could be improved. Surface characterization should be discussed. Why author didn't perform visualuzation of ASP in the macro scale and PP in the same scale as ASP and WPS? It should be explain why the same material ASP in the same scale 10um (figure 1 b,c) looks totally different.

Author Response

For the PP microporous membrane, microporous are usually only a few tens of nanometers wide. The structure of the microporous membrane is not clear at small magnification. So the magnification of PP microporous membrane is 20000, the scale bar length is 2 μm. Figure 1(b) is the surface of APS, and figure 1(c) is the sectional morphology of APS. In figure 1(c) we also can see the PP separator. The microporous in the PP diaphragm are barely visible. Figure 1 is illustrated in detail in this article

Reviewer 2 Report

Results revealed in abstract should be more quantitative. Lack of updated references in the introduction literature. Some recent literature should be cited in introduction part of the manuscript. The more details about PP porous membrane should be revealed in experimental part. Characterization: More details about the characterization parameters should be provided, e.g., the acceleration voltages used in SEM experiment and specimen preparation for the SEM examination, etc. How did authors calculate the pore size from SEM images? For the acetone and water coated PP porous membrane samples, the high magnification images should be provided to clearly understand the microstructure APS and WPS. As mentioned by author the better affinity between the separator and the liquid electrolyte shows enhanced electrolyte absorption/retention. So, the question is why WPS showed better electrolyte retention (%) when compared to APS, even though APS exhibited more affinity between the separator and the liquid electrolyte. The authors should discuss more explanation or cite any suitable The following recent literature on Lithium-ion batteries should be added. (i) https://doi.org/10.1016/j.matlet.2019.03.058  (ii) https://doi.org/10.1016/j.micromeso.2018.06.005  (iii) https://doi.org/10.1007/s11837-018-2888-y 

Author Response

The acceleration voltages used in SEM experiment and specimen preparation for the SEM examination, etc. How did authors calculate the pore size from SEM images?

Answer: The acceleration voltages used is SEM experiment has been added in manuscript. The pore size and coating layer thickness are calculated by the software of ImageJ. This information has been added in the manuscript.

Answer: The acceleration voltages used is SEM experiment has been added in manuscript. The pore size and coating layer thickness are calculated by the software of ImageJ. This information has been added in the manuscript.

For the acetone and water coated PP porous membrane samples, the high magnification images should be provided to clearly understand the microstructure APS and WPS.

Answer: A new figure has given in revised manuscript.

As mentioned by author the better affinity between the separator and the liquid electrolyte shows enhanced electrolyte absorption/retention. So, the question is why WPS showed better electrolyte retention (%) when compared to APS, even though APS exhibited more affinity between the separator and the liquid electrolyte.

Answer: Because the coating of WPS is denser than APS separator, its electrolyte absorption/retention rate is higher than that of the APS film. However, the infiltration process of electrolyte is time dependent. APS separator with a porous structure allow faster electrolyte infiltration due to the capillarity of the surface. Therefore, the affinity between the separator and the liquid electrolyte and the liquid electrolyte absorption/retention rate are two different processes, which are not directly related

The authors should discuss more explanation or cite any suitable The following recent literature on Lithium-ion batteries should be added. (i) https://doi.org/10.1016/j.matlet.2019.03.058, (ii) https://doi.org/10.1016/j.micromeso.2018.06.005, (iii) https://doi.org/10.1007/s11837-018-2888-y

Answer: The new references have been added to the revised manuscript.

Round 2

Reviewer 2 Report

Authors responded to my comments. Manuscript in its present form may consider for publication